# Using Aquatic Plant-Derived Biochars as Carbon Materials for the Negative Electrodes of Li-Ion Batteries

Andrey A. Belmesov [1,2], Alexander A. Glukhov [1,2], Ruslan R. Kayumov [2], Dmitry N. Podlesniy [2], Elena M. Latkovskaya [1], Maria A. Repina [1], Nikita P. Ivanov [3], Maxim V. Tsvetkov [1,2] and Oleg O. Shichalin [1,3,*]

[1] Sakhalin State University, 33, Kommunistichesky Ave., 693000 Yuzhno-Sakhalinsk, Russia
[2] Federal Research Center of Problems of Chemical Physics and Medicinal Chemistry, Russian Academy of Sciences, 1 Academician Semenov Av., 142432 Chernogolovka, Russia
[3] Nuclear Technology Laboratory, Department of Nuclear Technology, Institute of High Technologies and Advanced Materials, Far Eastern Federal University, 10 Ajax Bay, Russky Island, 690922 Vladivostok, Russia
* Correspondence: oleg_shich@mail.ru

**Abstract:** The current study focuses on the production of biochars derived from aquatic plants, specifically red seaweed *Ahnfeltia* and seagrass *Zostera* and *Ruppia*, found in brackish lagoons in the Sea of Okhotsk, Sakhalin Island. These biochars were obtained through a stepwise pyrolysis process conducted at temperatures of 500 and 700 °C. The characteristics of the biochars, including their elemental composition, specific surface area, and particle size distribution, were found to be influenced by both the type of biomass used and the pyrolysis temperature. The primary objective of this research was to investigate the potential of these biochars to be used as negative electrodes for lithium ion batteries. Among the various samples we tested, the biochar derived from the macroalgae *Ahnfeltia tobuchiensis*, produced at 700 °C, exhibited the highest carbon content (70 at%) and nitrogen content (>5 at%). The reversible capacity of this particular biochar was measured to be 391 mAh g$^{-1}$ during the initial cycles and remained relatively stable at around 300 mAh g$^{-1}$ after 25 cycles. These findings suggest that biochars derived from aquatic plants have the potential to be utilized as effective electrode materials in lithium ion batteries. The specific properties of the biochar, such as its elemental composition and surface area, play a significant role in determining its electrochemical performance. Further research and optimization of the pyrolysis conditions may lead to the development of biochar-based electrodes with improved capacity and cycling stability, thereby contributing to the advancement of sustainable and environmentally friendly energy storage systems.

**Keywords:** biochar; Li-ion battery; negative electrode; red seaweed *Ahnfeltia*; sea grasses *Zostera* and *Ruppia*; pyrolysis





## 1. Introduction

Currently, many different types of batteries based on various mobile ions are known, for example, H$^+$ [1], Li$^+$ [2,3], Na$^+$ [4], K$^+$ [5], Ag$^+$ [6], Cu$^+$ [7], NH$_4^+$ [8], Mg$^{2+}$ [9], Ca$^{2+}$ [10], Zn$^{2+}$ [11], F$^-$ [12], Cl$^-$ [13], etc. Lithium ion batteries (LIBs) are the most common due to their unique balance of properties—such as their capacity, current characteristics, availability, and charge safety. Their widespread use in various devices, the number of which is annually increasing, and the presence of some restrictions on the characteristics of materials used in them indicates the need to develop new systems for all three of their components: their negative and positive electrodes, divided by their electrolytes. The operation of LIBs is based on the movement of lithium ions between two electrodes through these electrolytes: during charging, Li$^+$ ions move from the positive electrode, rich in lithium, to the negative; and during discharge, on the contrary, they return from the negative to the positive electrode. Therefore, the negative electrode must have a specific structure to allow the introduction of a large number of lithium ions without a strong change in the volume of the

electrode material in order to not destroy the battery during charge–discharge processes. Today, one of the most common negative electrode materials used in commercial LIBs is graphite due to the space between its layers being sufficient to house Li$^+$ ions [14]. In particular, the low theoretical specific capacity (372 mAh g$^{-1}$) [15] and the low efficiency of graphite severely limit its use as a material for negative electrodes in LIBs [16]. Other carbon materials, such as carbon nanotubes or graphene, are often used to improve battery performance, but this has been found to be very expensive for mass-production [4].

Extensive research has been conducted to explore diverse materials and approaches for developing advanced negative electrode materials for LIBs. An analysis of the literature has shown that one of trends for improving LIB characteristics is the replacement of graphite with biochar. The increasing number of publications on this topic indicates the prospects of this strand of research. Biochar is a carbon-rich material produced during the pyrolysis process, which involves the thermochemical decomposition of a biomass. Depending on the type of biomass, the resulting carbon material may have a different structure, porosity, and composition [17]. An analysis of the literature shows that various types of biomass can be used to prepare biochar for use as a negative electrode material for metal ion batteries: coniferous and broadleaf wood [18,19], lignin [20], herbal plants (for example, bamboo [21] and cotton [22]), starch [23], coconut husks [24,25], banana peel [26,27], rice husks [28,29], grapefruit [30], hairs [31], various aquatic plants, [32] and even leather production waste [33]. The improvement in the electrochemical characteristics of biochar compared to those of graphite is explained by its large specific surface area (100–3000 m$^2$ g$^{-1}$), making it possible to achieve a specific capacity of 200–800 mAh g$^{-1}$ after 100 charge/discharge cycles [4,19,22,30,32]. In addition, such biochars contain a number of heteroatoms, for example, oxygen, sulfur, nitrogen, etc.; the presence of these can positively affect the structure and properties of the biomass-derived carbon material, while the pyrolysis temperature influences the amount of biochar and its morphology [34,35]. The preparation method (the temperature and duration of the pyrolysis, the heating rate, the catalysts used, the atmosphere of the pyrolysis process), composition, morphology, and their influence on the biochar's characteristics have been described in detail in the literature [4,17,36–38]. For example, increasing the pyrolysis temperature leads to a graphite-like structure of biochar and an increase in its carbon content, and a decrease in the porosity and content of its surface-active groups and heteroatoms. The rate of the pyrolysis also has a great influence: compared to slow pyrolysis, fast and flash pyrolysis reduces the carbon and oxygen content, the production time, and the biochar yield.

Macroalgae and seagrasses have emerged as promising feedstock for biochar production [4,39–46]. They are particularly relevant in coastal regions, including sea and lagoon coasts, salt marshes, and wetlands. The Sea of Okhotsk, located near southeastern Sakhalin Island in Russia, boasts a rich species diversity and biomass of aquatic flora. For instance, the coastal zone of the Sea of Okhotsk near the southeast of Sakhalin, including Aniva Bay, is home to a staggering 246 species of macroalgae [47].

During sampling in autumn 2021, it was observed that the mass of storm wrack composed of macrophytes on the littoral zone of this area can exceed 30 kg m$^{-2}$. This abundance of macrophytes presents an excellent opportunity for utilizing them as a feedstock for biochar production. Harnessing these aquatic resources can not only help manage the accumulation of biomass in coastal areas but can also provide a sustainable and renewable source for biochar production. By converting the macrophytes into biochar, it is possible to enhance carbon sequestration, improve soil quality, and explore various applications in agriculture, environmental remediation, and energy storage.

We sampled three macrophyte species from the coast of brackish lagoons (Sea of Okhotsk, Sakhalin). Red seaweed *Ahnfeltia fastigiate var. tobuchiensis* (hereinafter referred to as *Ahnfeltia tobuchiensis*) [48] and seagrass *Zostera marina* were sampled in the Busse lake littoral. On the littoral of the Tunaicha lake lagoon, several species of *Ruppia* (hereafter denoted as *Ruppia* sp.) seagrasses were selected. Seagrasses of the genus *Ruppia* are widespread in marine and brackish lagoons along the entire coast of the Sea of Okhotsk.

During storms, huge masses of *Ahnfeltia tobuchiensis* (AT, red seaweed), *Zostera marina* (ZM, seagrass) and *Ruppia* sp. (HAV, seagrass) are periodically casted ashore off the island coast; however, they are practically never recycled, although they are valuable raw materials. For example, red *Genus Ahnfeltia* contain a gel-forming polysaccharide agar (comprising up to 50% of their dry weight), so they are used for the production of gel-forming agents (agar, biofilms, etc.) [49].

Therefore, the use of such plants from storm drains may be promising due to their availability, as well as the processing simplicity. Recent investigations have reported the promise shown by biomass-derived carbon materials when used as a negative electrode in LIBs [4,39,50,51]; nonetheless, there is deficient knowledge on biomass characteristics/composition and the properties of biomass-derived carbon materials. Accordingly, the purpose of this work was to study the influence of the nature of aquatic plant biomasses from HAV, AT, and ZM and the pyrolysis temperature on the properties of the obtained biochar, regarding the possibility of using it as a negative electrode for LIBs.

## 2. Materials and Methods

As mentioned above, the sample of biomasses in our research comprised three types of aquatic plants—AT, ZM, and HAV—collected in the Busse lagoon of the Sea of Okhotsk. All materials were washed three times in tri-distilled water under ultrasound to remove sea salt, sand, and other possible impurities. Further, the biomaterial was dried in a Binder oven at 105 °C for 24 h. Biochars were obtained via pyrolysis in a quartz setup (Figure 1a), which was carried out in a nitrogen atmosphere. The HAV500, AT500, ZM500 samples were obtained via stepwise pyrolysis according to the scheme of temperature regime shown in Figure 1b (green line), and the HAV700, AT700, and ZM700 samples were obtained from parts of the grinded HAV500, AT500, ZM500, respectively (regime denoted with a red line). A minimum temperature of 500 °C was chosen to obtain a well-arranged, lattice-ordered carbon layer and ensure the complete decomposition of polysaccharides, proteins, lipids, and cellulose; and we chose a slow pyrolysis with a heating rate of <10 °C min$^{-1}$ as this leads to increased productivity [38]. The application of a stepwise pyrolysis in the field of biomass conversion is common and of importance to enhance the yield and quality of the target products [52,53]. According to [53], the stepwise pyrolysis method clearly demonstrates a higher char yield compared with the single-step pyrolysis.

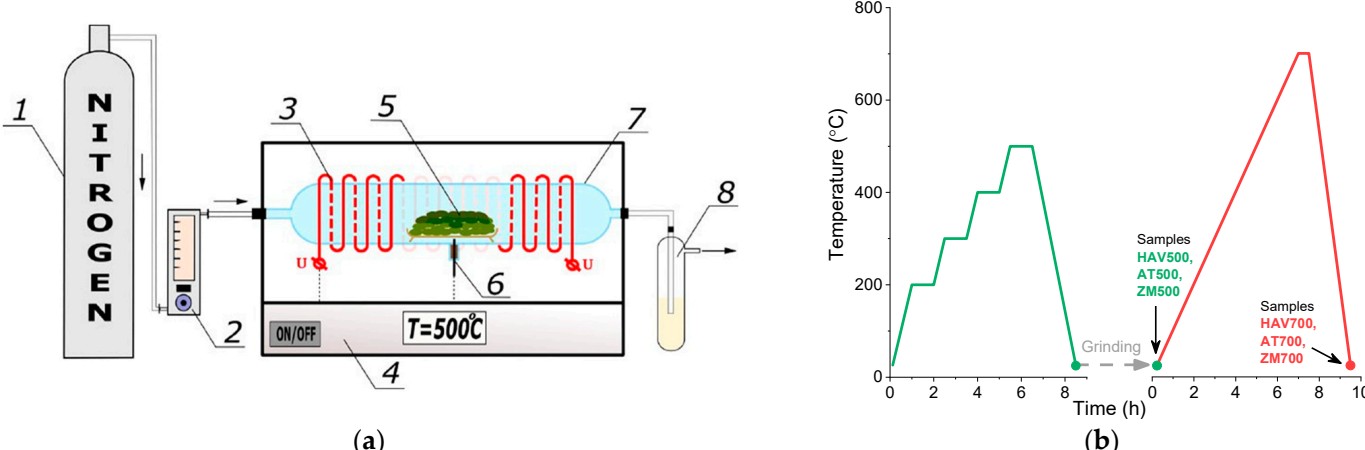

**Figure 1.** Scheme of biochar production via pyrolysis: (**a**) setup and (**b**) temperature regime. 1: Nitrogen reservoir, 2: electronic flow meter, 3: heating element, 4: control unit, 5: biomass samples, 6: K-type thermocouple, 7: cylindrical quartz reactor, 8: water seal.

Cooling was carried out in a free mode at room temperature overnight (the quartz reactor was removed from the oven without opening). The grinding of the biochar after

pyrolysis was carried out in cyclohexane in a planetary ball mill, "Pulverisette 6" (Fritsch), in a zirconium dioxide glass. The samples were stored in sealed containers.

Microphotographs of the samples with energy-dispersive X-ray (EDX) spectroscopy analyses were obtained using a scanning electron emission microscope (SEM), LEO SUPRA 25 (Carl Zeiss, Jena, Germany). The elemental analysis was performed on a CHNS Vario El Cube elemental composition analyzer (Elementar GmbH, Langenselbold, Germany) via combustion in an oxygen flow. The specific surface area (SSA) of the samples was determined via a method of nitrogen adsorption–desorption on a QUADRASORB SI instrument (Quantachrome Instruments, Boynton Beach, FL, USA). Samples were degassed for 3 h at 300 °C in a helium atmosphere. Particle size distribution with SSA estimation was determined with an "Analysette 22 Next" laser diffractometer (Fritsch, Pittsboro, NC, USA).

The electrochemical performance of half-cells with the studied biochars as their negative electrode was tested with two-electrode coin cells 2032 assembled in an Ar-filled glove box with $O_2$ and $H_2O$ contents of <0.1 ppm. The half-cells contained a composite biochar electrode, a lithium metal counter electrode (15.6 × 0.25 mm lithium disks from Gelon LIB), a commercial liquid electrolyte 1 M $LiPF_6$ in organic solvent (ethylene carbonate + dimethyl carbonate + diethyl carbonate; 1:1:1 by volume) (Gelon LIB, Dongguan, China), and a polypropylene separator with a thickness of 80 μm (Gelon LIB). The composite electrodes were prepared via doctor blade coating of the homogenized mixture of the active material (biochar), a conductive additive (acetylene black Super P, provided by Gelon LIB), with polyvinylidene fluoride (PVDF) as a binder and N-methyl-2-pyrrolidone as a solvent onto a copper foil. The weight ratio of the dry components biochar–acetylene black–PVDF in the composite electrode mixture was 80:10:10. The electrode layer was subsequently dried at 60 °C for 1 h, and then compacted under a rolling press at 120 °C. Electrodes were cut in the form of discs, 13 mm in diameter, and dried under a vacuum at 120 °C for 15 h. The biochar loading was 10 mg $cm^{-2}$. For comparison, a half-cell with a graphite-based («TOB-Graphite-R», Gelon LIB) composite electrode was prepared.

Both the cycling performance and the capability rate of the half-cells were examined at room temperature using galvanostatic charge–discharge curves, measured with P-40X and P-45X potentiostat ("Elins" LLC, Marietta, GA, USA) in the voltage ranges of 0–1.5 V vs. $Li^0/Li^+$. The operating current density was 36 mA $g^{-1}$; the cell with the highest capacity values was tested under charge–discharging rates of 50–400 mA $g^{-1}$.

### 3. Results and Discussion

According to our SEM images (Figure 2), the biochar particles in all samples have a lamellar shape, with differences in their thicknesses and sizes of up to tens of microns. Among the biochar particles, there were a very small portion of particles with a complex porous structure (Figure 2), characteristic of diatoms [39,54]; these particles have a lot of regular oblong pores (0.2 μm wide and 0.5–0.7 μm long). The same pores were observed on the surface of biochars obtained from *Cladophora glomerata* in [39,54]. The authors of these works associate the formation of these pores with cracking and volatilization during the pyrolysis process.

The data obtained from the CHNS and EDX analyses (Table 1) are in good agreement with the data given in the recent reported work [4]: with increases in the pyrolysis temperature, the content of carbon and sulfur increases by 1.5–4.6 at%, meanwhile the content of the surface groups in the biochar samples decreases. The greatest influence of pyrolysis temperature on the carbon content was observed for the ZM samples. It can be assumed that the best material for LIB negative electrodes would be biochar obtained from AT, due to the fact that it has the maximum content of carbon (67–70 at%) and nitrogen (>5 at%).

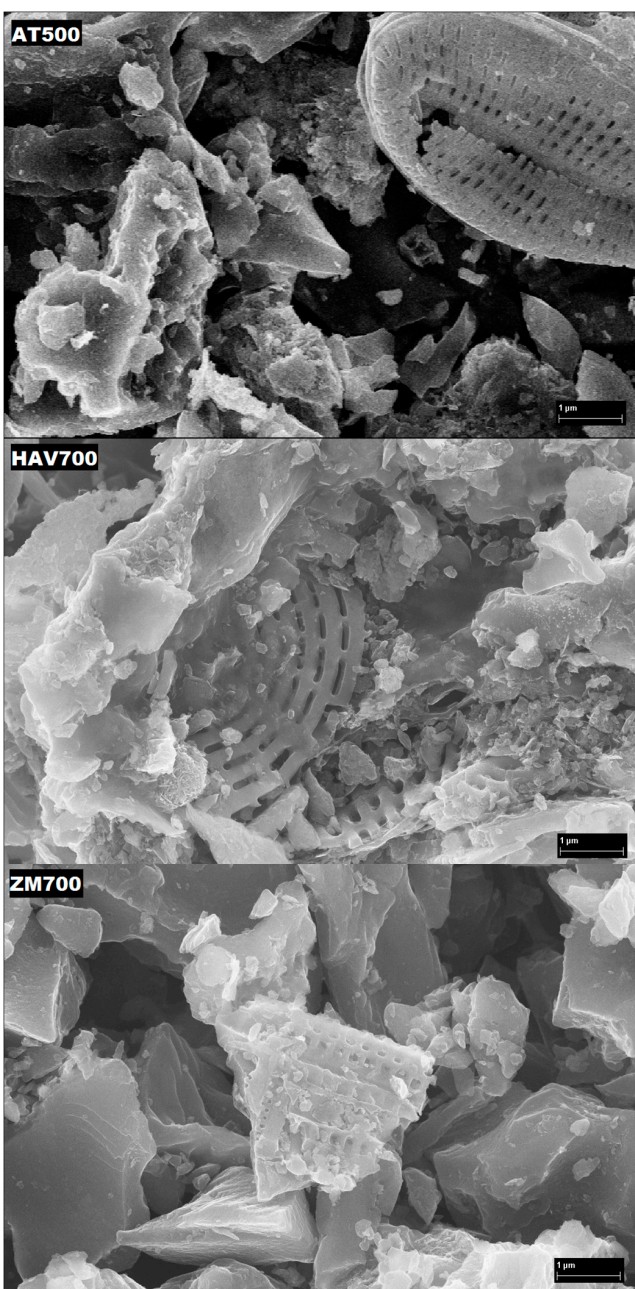

**Figure 2.** SEM images of the studied biochars.

**Table 1.** Data on the elemental composition of the studied biochars from CHNS/EDX analyses (at%).

| Element | HAV500 | HAV700 | AT500 | AT700 | ZM500 | ZM700 |
|---|---|---|---|---|---|---|
| C | 55.0/54.6 | 56.5/68.0 | 67.0/63.4 | 69.5/79.9 | 60.1/66.9 | 64.7/69.7 |
| H | 2.5/− | 1.7/− | 2.5/− | 2.1/− | 2.6/− | 1.9/− |
| N | 3.0/− | 3.0/− | 5.8/− | 5.5/− | 2.4/− | 2.4/− |
| S | 1.0/2.3 | 1.5/0.7 | 0.9/1.3 | 1.6/1.1 | 0.8/1.2 | 1.2/0.8 |
| O | −/28.3 | −/24.1 | −/28.9 | −/12.3 | −/18.9 | −/17.2 |
| Other elements (Mg, K, Ca, Al, Na, Si, Cl, Br) | −/14.8 | −/7.2 | −/6.4 | −/6.7 | −/13.0 | −/12.3 |

The convergence of the results between these two methods is not high due to the limitations of the EDX method for the determination of light elements. It is fundamentally impossible to determine the presence of hydrogen using this method, as the nitrogen peak cannot be separated from the overlapping carbon peak, which is 10–20 times more intense; the aluminum and bromine peaks were also overlapped (Figure 3). In addition, the EDX data indicate the incorrectness of attributing entire the remaining mass in elemental analyses to oxygen, since samples can also contain impurities of other elements (Mg, K, Ca, Al, Na, Si, Cl, Br).

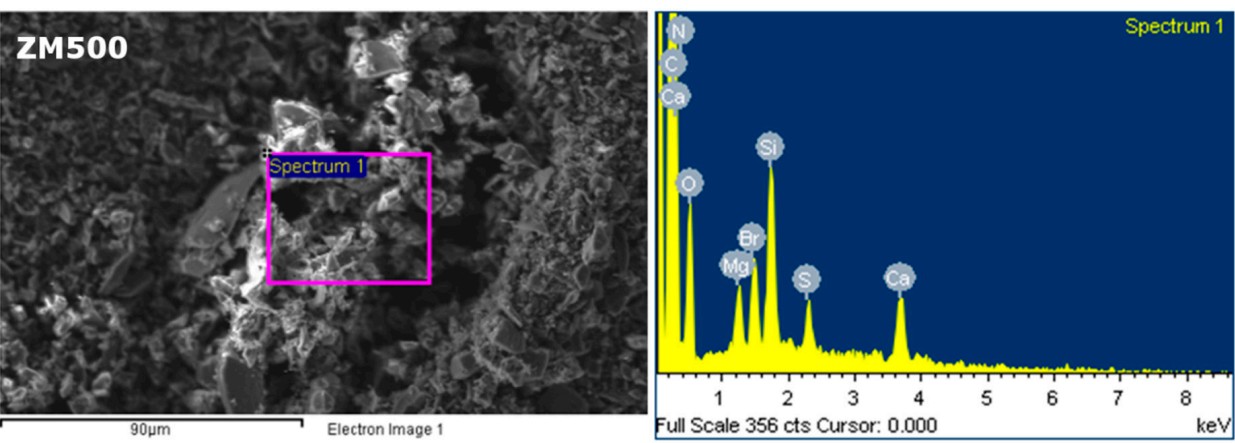

**Figure 3.** SEM images and corresponding EDX spectra of the ZM500 sample.

Biochar particles have a wide size distribution, from 0.2 to 70 μm. An analysis of the particle size showed that the HAV samples have a trimodal distribution, while the AT and ZM samples have a bimodal distribution, which can be perfectly described with a lognormal function (Figure 4):

$$\varphi = \frac{A}{\sqrt{2\pi}\omega d} exp\left[-\frac{(\ln d - \mu)^2}{2w^2}\right], \tag{1}$$

where $\varphi$ is the particle fraction, $A$ is the peak area, $\omega$ and $\mu$ are calculated parameters, and $d$ is the particle diameter.

The smallest particle size (<20 μm) is found in the ZM samples, with a mode of 2 μm. For this type of biochar, upon increasing the pyrolysis temperature, a noticeable increase in the number of large particles was observed. For the other samples, the effect of the pyrolysis temperature on the particle size is almost insignificant.

Furthermore, nitrogen adsorption–desorption isotherms were analyzed for the biochars under investigation, and these results are depicted in Figure 5a. All samples exhibited hysteresis loops in their isotherms, indicating the presence of mesopores and macropores. Based on the IUPAC classification [55], these isotherms were identified as type IV. At low pressures ($p/p_o < 0.1$), there was a sharp increase in volume (V), which can be attributed to the high adsorption potential of micropores with sizes smaller than 2 nm. Furthermore, all biochars displayed a H4 type of hysteresis loop across the entire range of relative pressures ($p/p_o$). The extended hysteresis observed at the minimum relative pressure is a result of the adsorbate being retained in narrow slit-like micropores. Our application of the Density Functional Theory (DFT) method allowed for the determination of the differential pore size distributions. It was found that all samples exclusively contained micropores with sizes smaller than 1 nm. No mesopores or macropores were observed, regardless of the type of biomass or the pyrolysis temperature (Figure 5b, Table 2).

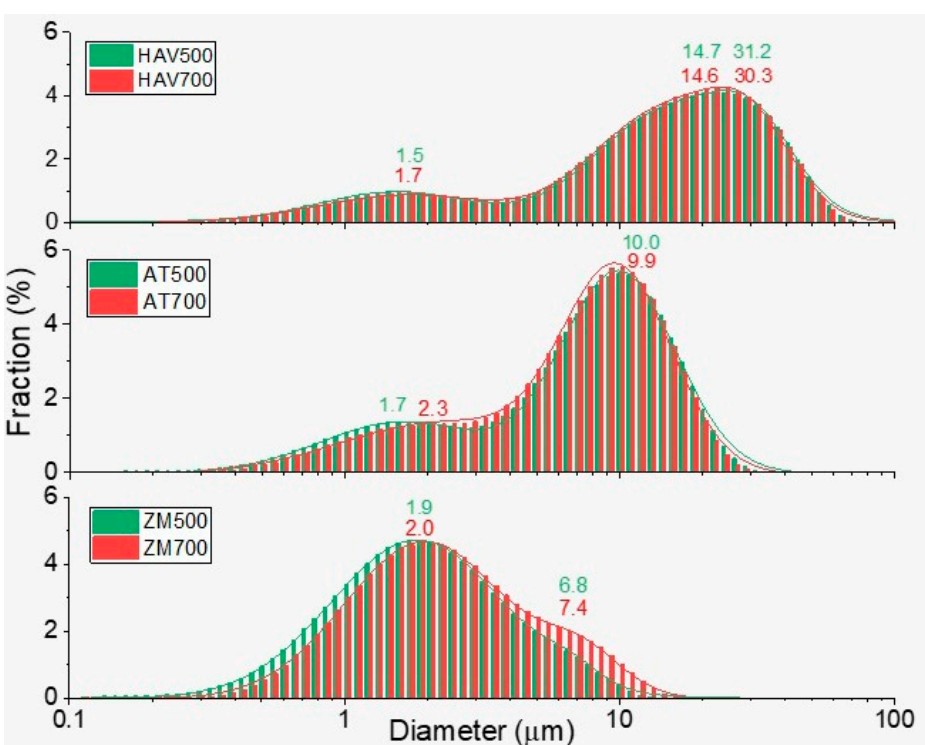

**Figure 4.** Particle size distributions of the biochar (lines and numbers show the approximation of histograms using Equation (1)).

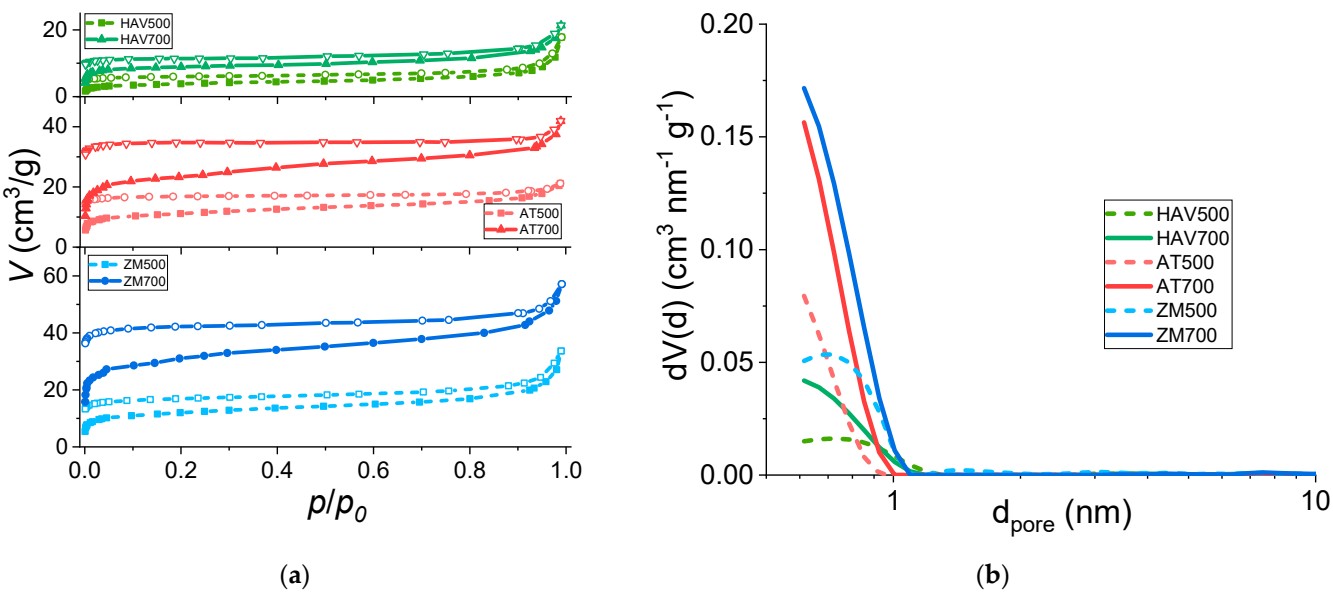

**Figure 5.** (**a**) Nitrogen adsorption (solid points)–desorption (open points) isotherms, and (**b**) differential pore size distribution, for the studied biochars.

Further analysis of the studied biochars revealed that they predominantly possess a micro-porous structure, characterized by a network of small-sized pores. This microporous nature suggests a high potential for adsorption and surface reactivity, making these biochars well-suited for diverse applications such as environmental remediation, water treatment, and energy storage.

In particular, among the biochars investigated here, the pore size distributions exhibited their maximum at approximately 0.7 nm for HAV500 and ZM500. On the other hand,

the most probable pore size for the remaining materials was found to be below 0.7 nm. This variation in pore size distribution indicates subtle differences in the porous characteristics of the biochars, which can influence their adsorption capacity and selectivity for different target substances.

**Table 2.** Laser diffraction and nitrogen adsorption–desorption data processed via different methods.

| Sample | $S_{LD}$ (m² g⁻¹) | $S_{BET}$ (m² g⁻¹) | DR | | T-Plot | | | DFT | |
|---|---|---|---|---|---|---|---|---|---|
| | | | $V_{micro}$ (cm³ g⁻¹) | $S_{micro}$ (m² g⁻¹) | $V_{micro}$ (cm³ g⁻¹) | $S_{micro}$ (m² g⁻¹) | $S_{surf}$ (m² g⁻¹) | $V$ (cm³ g⁻¹) | $d$ (nm) |
| HAV500 | 0.7 | 13.6 | 0.006 | 16.1 | 0.002 | 4.74 | 8.90 | 0.017 | 0.72 |
| HAV700 | 0.6 | 34.3 | 0.014 | 39.7 | 0.009 | 23.6 | 10.7 | 0.025 | 0.61 |
| AT500 | 0.9 | 41.0 | 0.017 | 48.5 | 0.009 | 21.8 | 19.2 | 0.028 | 0.61 |
| AT700 | 0.8 | 88.4 | 0.037 | 103.1 | 0.022 | 55.5 | 32.9 | 0.053 | 0.61 |
| ZM500 | 2.1 | 44.0 | 0.017 | 47.9 | 0.009 | 23.1 | 20.9 | 0.039 | 0.72 |
| ZM700 | 1.8 | 111.9 | 0.055 | 153.6 | 0.026 | 61.6 | 50.3 | 0.072 | 0.61 |

Table 2 compares the SSA values obtained via the laser diffraction method ($S_{LD}$) and determined with the Brunauer–Emmett–Teller (BET) method ($S_{BET}$), according to the following equation [55]:

$$V = \frac{V_m C \frac{p}{p_0}}{\left(1 - \frac{p}{p_0}\right)\left(1 + (C-1)\frac{p}{p_0}\right)}, \tag{2}$$

where $V$ is the volume of gas adsorbed at a standard temperature and pressure, $V_m$ is the volume of gas adsorbed at a standard temperature and pressure to produce an apparent monolayer on the sample surface, $C$ is the constant related to the energy of interaction with the surface, $p$ is the partial vapor pressure of adsorbate gas in equilibrium with the surface at $-196\,°C$, and $p_0$ is the saturated pressure of adsorbate gas. BET graphs are linear in coordinates $\frac{1}{V\left(\frac{p_0}{p}-1\right)}$ vs. $p/p_0$ in the low-pressure region $p/p_0 = 0.01$–$0.1$. It is apparent that the $S_{BET}$ values are ten times higher than the $S_{LD}$ values. This suggests that the particles have a complex branched-surface structure, which is not taken into account for calculations in the laser diffraction method. The presence of a developed structure was confirmed by the nitrogen adsorption–desorption data (Figure 5b, Table 2) and SEM images (Figure 2).

For the microporous samples, calculating the $S_{BET}$ value using the standard method may lead to erroneous results. Due to the fact that adsorption in micropores has its own characteristics (it occurs not on the surface of pores, but in its entire volume), the Dubinin–Radushkevich (DR) theory [56], the theory of volumetric filling of micropores (a special case of the Dubinin–Astakhov theory), was applied to describe it:

$$V = V_0 \exp\left[-\left(\frac{RT}{E}\ln\left(\frac{p}{p_0}\right)\right)^2\right] \tag{3}$$

where $R$ is the universal gas constant, $T$ is the absolute temperature, and $E$ is the characteristic adsorption free energy. To determine the types of pores in the studied material, as well as to separate adsorption in pores of various sizes, the T-plot–De Boer method was used (regarding the dependence of the adsorption value on the thickness of adsorption film in a standard sample). In addition to information about the presence of different types of pores in a sample, the T-plot method allows us to calculate the true volume of micropores ($V_{micro}$), while excluding adsorption on the outer surface.

In general, the SSA value determined using these three methods coincides (Table 2) and increases by more than two times with the increase in pyrolysis temperature, which correlates well with the data reported in the literature [4,57]. The main augmentation occurs due to an increase in the sample's microporosity, which is probably due to the removal of oxygen-containing groups and other groups with the formation of pores in this

place. As the pyrolysis temperature increases, unstable components of algae (for example, polysaccharides, proteins, and lipids) tend to decompose into volatile compounds such as acetic acid, ammonia, methanol, $CO_2$, $CO$, and $H_2$. The release of gaseous substances facilitates the unblocking of pore channels to form more porous structures in biochars [51]. The biochars obtained from HAV have the lowest SSA: even after pyrolysis at 700 °C, their SSA does not exceed 40 m$^2$ g$^{-1}$. For ZM700, $S_{BET}$ = 112 m$^2$ g$^{-1}$. At the same time, the external SSA ($S_{surf}$) accounts for a little less than half of the entire SSA. Comparing this with data from the T-plot method, it can be seen that volume of micropores ($V_{micro}$) calculated using the DR and DFT methods appears to be overestimated, which means that the contribution of adsorption on the outer surface is significant.

Our testing of the obtained biochars as negative electrodes in half-cells revealed their initial capacity to be in the range of 150–450 mAh g$^{-1}$, which decreased to 100–300 mAh g$^{-1}$ by the 25th cycle (Figure 6a,b). For comparison, the theoretical capacity of graphite is 372 mAh g$^{-1}$ [32,50]. This decrease in specific capacity during cycling can be attributed to the presence of numerous surface groups that bind with lithium, resulting in a low Coulombic efficiency during the initial cycles (Figure 6c) [51]. The low Coulombic efficiency in the first cycle, ranging from 30% to 44%, is likely associated with the formation of a solid electrolyte interface (SEI) on the surface of the biochars. Considering the extensive surface area of the biochars, a significant amount of supplied electricity is consumed in the formation of this SEI [4].

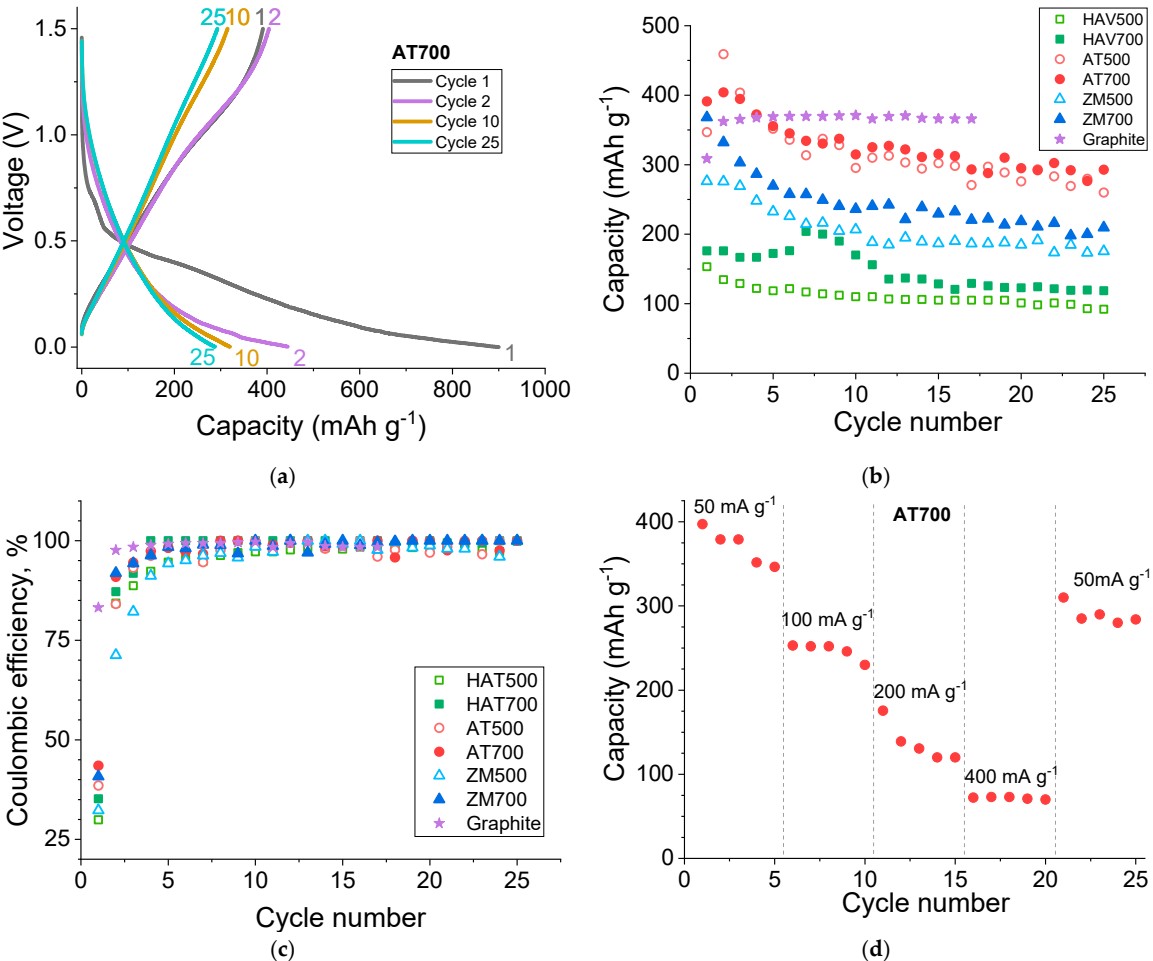

**Figure 6.** (**a**) Galvanostatic charge and discharge curves for the cell with AT700; (**b**) cycling performance of the cells with the studied biochars; (**c**) Coulombic efficiency at a current density of 36 mA g$^{-1}$; (**d**) cycling performance of the cell with AT700 at different charge–discharge rates.

It should be noted that the formation of the SEI occurs primarily in the first cycle, as is indicated by the shape of the lithium intercalation curve. The shape of the intercalation curve in the first cycle differs significantly from that of the second cycle, which remains relatively unchanged. The shape of the lithium deintercalation curve for the first two cycles is almost identical, but changes with further cycling [32,39]. The decrease in capacity within the potential range of 0.3–1.3 V vs. $Li^0/Li^+$ can be attributed to the reversible interaction of $Li^+$ with the porous structure of the biochars, resulting in an increase in capacity similar to that observed in hard carbon and soft carbon materials [58]. However, the capacity within the potential range of 0–0.3 V vs. $Li^0/Li^+$ (related to the intercalation/deintercalation of lithium between carbon layers) remains relatively unchanged. It appears that, before the 10th cycle (the end of the rapid decrease in capacity), with each cycle, a portion of the surface groups on the biochars is irreversibly reduced due to the lithium, leading to a decrease in their capacity and Coulombic efficiency during these cycles. It is also worth noting that the Coulombic efficiency of hard carbon and soft carbon materials is known to deviate from 100% [38].

For all of the biochar samples obtained at higher temperature (700 °C), the Coulombic efficiency was higher in the first cycles. This means that the influence of the surface groups on the irreversible capacity is greater than that of the increased sample surface area. The SSA value does not have a direct effect on the residual battery capacity after cycling: sample ZM700, which has the maximum SSA among the studied biochars, has average residual capacity values. These results coincide well with the data reported in the literature. For example, the residual capacity of an LIB based on biochar from cherry pits (SSA > 1600 $m^2$ $g^{-1}$) is 225 mAh $g^{-1}$ after 200 cycles [59], while the use of graphite-like materials from banana peels (SSA = 217 $m^2$ $g^{-1}$) makes it possible to obtain devices with 800 mAh $g^{-1}$ after 400 cycles [60].

AT-derived materials have the highest capacity. This is probably due to the fact that such biochars contain a larger amount of nitrogen and the smallest amount of metal impurities (Table 1), which has a positive effect on the capacity characteristics of carbon materials [4,61–63]. The presence of such heteroatoms as N, S, B, and P allows one to increase a material's electrical conductivity. On the other hand, a reduction in metal impurities improves the stability of the material for use as a negative electrode in metal ion batteries. The bulk density of the AT700 biochar was 0.64 ± 0.02 g $cm^{-3}$, and the density of the composite electrode material with it was 0.72 ± 0.04 g $cm^{-3}$.

In general, each specific capacity value of the studied samples correlates with their carbon content: with an increase in the C content, the specific capacity increases (Figure 6b). The capacity of the obtained samples exceeds the capacity of graphite, but only in the first cycles. However, the undeniable advantage of the samples obtained in this study is their somewhat-higher Coulomb efficiency, which, after the 10th cycle, is close to 100% (Figure 6c). As in the case of the traditional negative electrode material (graphite), the successful application of the resulting biochars requires the use of additional additives in the electrolyte, the optimization of the binder, and additional pre-treatments of the material, which can significantly improve the cycling parameters [64,65].

The AT700 sample displayed a maximum capacity of 293 mAh $g^{-1}$ at the 25th cycle and was tested under various charge–discharge rates. It can be seen that as the current density increases to 400 mA $g^{-1}$, the specific capacity of this sample decreases stepwise to 73 mAh $g^{-1}$ (Figure 6d). When the current density was rolled back to 50 mAh $g^{-1}$, this cell lost 18% of its capacity, compared to the 5th cycle at that current density. The fact that this cell loses its initial capacity can be explained by the partial degradation of the biochar. According to [4,19,22,30,32], the capacity of LIBs with biochar material of different natures is in the range of 200 to 800 mAh $g^{-1}$ after 30–700 cycles. It is worth noting that the biochar samples presented in these reviews were subjected to significant pre-treatments, for example, hydrothermal synthesis, and acid and/or alkaline treatment. The materials studied in this work were tested in LIBs without any modification, which was needed in

order to avoid a decrease in their capacity during device cycling for the stable operation of the LIBs.

The presence of well-defined micropores suggests the potential for efficient adsorption of molecules and ions within this specific range. This property can be advantageous for applications where the removal of contaminants or the storage of small molecules is desired. By understanding and manipulating these pore characteristics, the biochars can be adapted to meet the specific requirements of various applications, thus maximizing their effectiveness and functionality (for example, for novel systems like reserve lithium ion batteries [66]).

## 4. Conclusions

Aquatic plant-derived biochars were produced through stepwise pyrolysis at temperatures of 500 and 700 °C, resulting in particles ranging in size from 0.2 to 70 μm. These biochars exhibited a complex branched surface with a high microporosity and a specific surface area ranging from 14 to 112 $m^2 g^{-1}$. The main component of these biochars was carbon, accounting for 55 to 70 atomic percent, while oxygen comprised 12 to 29 atomic percent. Other elements, such as nitrogen, hydrogen, sulfur, calcium, and potassium, were present in low amounts. Increasing the temperature of the pyrolysis process had a minimal impact on the particle size of the biochars, but significantly influenced their elemental composition and surface area. When evaluated as negative electrode materials for lithium ion batteries (LIBs), the biochars exhibited a capacity of 150–400 mAh $g^{-1}$ during the first cycle and 100–300 mAh $g^{-1}$ by the 25th cycle. Among the biochars, those derived from aquatic plants showed the highest capacity, likely due to their composition containing a higher proportion of carbon and nitrogen, as well as their optimal particle size and porous structure. These findings suggest that aquatic plant-derived biochars have the potential to be used as electrode materials for LIBs, offering promising properties for energy storage applications.

**Author Contributions:** Methodology, A.A.B., A.A.G., R.R.K., N.P.I., D.N.P. and E.M.L.; formal analysis, A.A.B., A.A.G., R.R.K., M.A.R. and O.O.S.; investigation, A.A.B., N.P.I., A.A.G., D.N.P. and M.V.T.; data curation, R.R.K., M.V.T. and O.O.S.; writing—original draft preparation, A.A.B., A.A.G. and R.R.K.; writing—review and editing, A.A.B., A.A.G., R.R.K., D.N.P. and O.O.S.; supervision, M.V.T. All authors have read and agreed to the published version of the manuscript.

**Funding:** This work was carried out in accordance with the State Assignment of Sakhalin State University (registration No. 122012500156-1).

**Data Availability Statement:** Data are contained within the article.

**Acknowledgments:** This work has been performed using equipment from the Federal Research Center for Problems of Chemical Physics and Medicinal Chemistry, of the Russian Academy of Sciences.

**Conflicts of Interest:** The authors declare no conflict of interest.

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
