# Peer review of "Using Aquatic Plant-Derived Biochars as Carbon Materials for the Negative Electrodes of Li-Ion Batteries"

_coatings, doi:10.3390/coatings13122075_

Round 1

Reviewer 1 Report

Comments and Suggestions for Authors

The proposed work deals with the application of biochar derived from aquatic plant in Li-ion batteries. The work appears well written, the results are satisfactorily discussed, and the materials employed has been well characterized. I would only suggest further characterization of the spent materials. 

I believe the manuscript can be accepted after minor revision.

Here are few comments:

·      Lines 68-78. Some references should be added to these statements. You could find something at  https://doi.org/10.1016/j.fuel.2020.119759 and https://doi.org/10.1016/j.jaap.2021.105337.

·      For figures 6 a and d, I suggest the authors to include within each graph that they are referred to sample AT700, to give more readability despite this information is already included in the caption. 

·      No characterization of the samples was performed after the 25th cycle. Do the authors expect a change in the textural properties of the material?

Author Response

Re: «Aquatic plant-derived biochars as carbon materials for negative electrode of Li-ion batteries» by A. A. Belmesov and et al. (Manuscript Number: coatings-2757514)

Dear Editors, dear Reviewers,

We deeply appreciate the time you spent reviewing our paper and the valuable recommendations you made. All the comments are taken into account and corresponding changes are made to the manuscript’s body text. Detailed point-by-point answers are presented below.

On behalf of co-authors,

Oleg Shichalin, Researcher, Ph.D.

Response to Reviewers

Reviewer #1:

The proposed work deals with the application of biochar derived from aquatic plant in Li-ion batteries. The work appears well written, the results are satisfactorily discussed, and the materials employed has been well characterized. I would only suggest further characterization of the spent materials. 

I believe the manuscript can be accepted after minor revision.

Here are few comments:

Comment #1

Lines 68-78. Some references should be added to these statements. You could find something at  https://doi.org/10.1016/j.fuel.2020.119759 and https://doi.org/10.1016/j.jaap.2021.105337.

Response #1

Thank you, references have been added. (Line 74).

Comment #2

For figures 6 a and d, I suggest the authors to include within each graph that they are referred to sample AT700, to give more readability despite this information is already included in the caption. 

Response #2

Thank you for your request. . Paper was revised for such type of failures. Specifically Fig.6a and 6d were modified.

Comment #3

No characterization of the samples was performed after the 25th cycle. Do the authors expect a change in the textural properties of the material?

Response #3

Thank you for your request. A change in the appearance of the samples after cycling is expected due to the formation of SEI on the surface of material; no other significant changes are expected, since there is no change in capacity several times from the original one.

Reviewer 2 Report

Comments and Suggestions for Authors

The purpose of this paper is to study the properties of aquatic plant biomass from HAV, AT and ZM and the influence of pyrolysis temperature on the properties of biochar, so as to explore the possibility of using biochar as the negative electrode of LIB. The research is interesting. However, major revisions are needed and the following comments are made.

1)    Please pay attention to the typos. For example, “plant-derivedbiocharsas” in the title should be revised.

2)    Various electrochemical energy storage devices have been developed in recently years. What are the advantages of Li ion batteries compared to other batteries? Please add a short comparison by citing some typical references, such as Journal of Alloys and Compounds 2022, 903, 163824; Rare Metals 2022, 41 (10), 3432-3445.

3)    The pyrolysis temperature of the experimental part is only 500 °C and 700 °C, and the comparison data is not enough. Different temperatures should be added to find out the optimal pyrolysis temperature.

4)    The Y axial of Figure 5 (a) should be labelled.

5)    EIS spectra are suggested to be added to reveal the resistance of different electrodes. Please refer and cite Chemical Engineering Journal 2023, 458, 141381.

6)    Please pay attention to the writing of units. For example, “mA/g” and “mA g-1” should be written in the same way.

7)    Only the performance test in the half battery was done, and it is hoped to add the performance test in the whole battery to support the viewpoint of the paper.

8)    What cause the capacity difference of biochar? Please give some explainations.

9)    The stability is not good. Please give some reasons.

Comments on the Quality of English Language

Moderate editing of English language is required.

Author Response

Re: «Aquatic plant-derived biochars as carbon materials for negative electrode of Li-ion batteries» by A. A. Belmesov and et al. (Manuscript Number: coatings-2757514)

Dear Editors, dear Reviewers,

We deeply appreciate the time you spent reviewing our paper and the valuable recommendations you made. All the comments are taken into account and corresponding changes are made to the manuscript’s body text. Detailed point-by-point answers are presented below.

On behalf of co-authors,

Oleg Shichalin, Researcher, Ph.D.

Response to Reviewers

Reviewer #2:

The purpose of this paper is to study the properties of aquatic plant biomass from HAV, AT and ZM and the influence of pyrolysis temperature on the properties of biochar, so as to explore the possibility of using biochar as the negative electrode of LIB. The research is interesting. However, major revisions are needed and the following comments are made.

Comment #1

Please pay attention to the typos. For example, “plant-derived biochars as” in the title should be revised.

Response #1

Thank you for your request, paper was revised for typos.

Comment #2

Various electrochemical energy storage devices have been developed in recently years. What are the advantages of Li ion batteries compared to other batteries? Please add a short comparison by citing some typical references, such as Journal of Alloys and Compounds 2022, 903, 163824; Rare Metals 2022, 41 (10), 3432-3445.

Response #2

Thank you for your request. We included the necessary references (line 37) in the article: “Currently, many different types of batteries based on various mobile ions are known, for example, H+ [1], Li+ [2, 3], Na+ [4], K+[5], Ag+[6], Cu+[7], NH4+[8], Mg2+[9], Ca2+[10], Zn2+[11], F-[12], Cl-[13], etc. Lithium ion batteries (LIBs) are the most common due to the unique balance of properties - capacity, current characteristics, availability, and charge safety." (Lines 36-40).

Comment #3

The pyrolysis temperature of the experimental part is only 500 °C and 700 °C, and the comparison data is not enough. Different temperatures should be added to find out the optimal pyrolysis temperature.

Response #3

Thank you for your interest in our work on biochar. It is important to note that this is our first work in this field, and we made efforts to select a temperature that matched the natural carbon carriers. However, we only used temperatures of 500 °C and 700 °C for the experimental part of the study. Experiments at different temperatures need to be added to determine the optimal pyrolysis temperature. We greatly appreciate your support and would appreciate the opportunity to further develop our laboratory.

Comment #4

The Y axial of Figure 5 (a) should be labelled.

Response #4

Thank you for your request, The Y axial of Figure 5(a) has been labelled.

Comment #5

EIS spectra are suggested to be added to reveal the resistance of different electrodes. Please refer and cite Chemical Engineering Journal 2023, 458, 141381.

Response #5

Thank you for your request. The EIS of the resulting half-cells have a more complex, significantly different appearance from the EIS given in the article “Chemical Engineering Journal 2023, 458, 141381” (In our paper this study was cited as [8]). Since this article shows a significantly different electrochemical system, we consider that it is not correct to interpret the EIS related to cells with biochar. And since they have a much more complex appearance, it is impossible to speak clearly about the resistance of the electrodes without additional research. At the same time, carbon black was added to the electrode to create the same conductive structure in the electrodes, so studying the conductivity of the electrodes does not make sense.

Comment #6

Please pay attention to the writing of units. For example, “mA/g” and “mA g-1” should be written in the same way.

Response #6

Thank you for your request. Paper was revised for such type of failures. Specifically Fig.6a and 6d were modified.

Comment #7

Only the performance test in the half battery was done, and it is hoped to add the performance test in the whole battery to support the viewpoint of the paper.

Response #7

Thank you for bringing up this important point. In our current study, we conducted the performance test using only half of the battery. However, we acknowledge the significance of including a performance test for the entire battery to further support the viewpoints presented in the paper. We appreciate your input and will take it into consideration for future research.

Comment #8

What cause the capacity difference of biochar? Please give some explainations.

Response #8

Thank you for your request. The difference in the capacity of biochars is due to differences in their composition and morphology, and these two parameters are highly dependent on the source plants. Since the pyrolysis temperature is not high, the porous matrix in biochars is preserved, and the capacity of materials such as soft carbon ([38] and [58]) strongly depends on it. The capacity also depends on the content of heteroatoms in the carbon structure, which varies in the samples. Thus, the combination of these two parameters determines such difference of materials.

Comment #9

The stability is not good. Please give some reasons.

Response #9

Thank you for your request. Low stability is typical for soft carbon materials. It is also due to the presence of a large number of surface oxygen (and other) groups. We deliberately did not introduce additional pre-treatments that may change the composition of the material in order to determine the characteristics of the materials in their original state.

Reviewer 3 Report

Comments and Suggestions for Authors

The paper by Andrey A. Belmesov et al. reported a detailed comparison on the properties and performance of biochars derived from different aquatic plants as negative electrode of Li-ion batteries. It provided quantitative insight of specific properties of the biochar in determining its electrochemical performance. There are few issues needs to be addressed specifically before consideration for publication.

1. The experimental procedures of producing biochar (HAV500, AT500, HAV700, etc.) are still confusing. It is recommended to add few sentences somewhere between line 123 and 133 to clearly explain the these labels refer to different pyrolysis temperature and the HAV700, etc. are produced under step-wise pyrolysis. Also the labels should be described in the caption of Figure 1.

2. SEM images of studied biochars (Figure 2) look confusing to me. It seems the right-column images are zoomed-in images. It is highly suggested to re-construct the images and use same scale bars. Also it should be cautious of the description in the main text (line 168-171). As to me, I couldn't tell it is porous from the images.

3. The authors should be cautious of the discussion and conclusion (line 361-362). Although the experimental data shows more significant difference on the C/N content, it may not be the direct reason. It could be a comprehensive results from many physical properties.  The micropore size match to the Li+, the Li+ conductivity, etc. may matter. 

Comments on the Quality of English Language

Moderate edits are still needed.

Author Response

Re: «Aquatic plant-derived biochars as carbon materials for negative electrode of Li-ion batteries» by A. A. Belmesov and et al. (Manuscript Number: coatings-2757514)

Dear Editors, dear Reviewers,

We deeply appreciate the time you spent reviewing our paper and the valuable recommendations you made. All the comments are taken into account and corresponding changes are made to the manuscript’s body text. Detailed point-by-point answers are presented below.

On behalf of co-authors,

Oleg Shichalin, Researcher, Ph.D.

Response to Reviewers

Reviewer #3:

The paper by Andrey A. Belmesov et al. reported a detailed comparison on the properties and performance of biochars derived from different aquatic plants as negative electrode of Li-ion batteries. It provided quantitative insight of specific properties of the biochar in determining its electrochemical performance. There are few issues needs to be addressed specifically before consideration for publication.

Comment #1

The experimental procedures of producing biochar (HAV500, AT500, HAV700, etc.) are still confusing. It is recommended to add few sentences somewhere between line 123 and 133 to clearly explain the these labels refer to different pyrolysis temperature and the HAV700, etc. are produced under step-wise pyrolysis. Also the labels should be described in the caption of Figure 1.

Response #1

Thanks to the reviewer for the question.

We modified text as follows: “HAV500, AT500, ZM500 samples were obtained by stepwise pyrolysis according to the scheme of temperature regime shown in Figure 1b (green line), and HAV700, AT700, ZM700 samples were obtained from part of the grinded HAV500, AT500, ZM500, respectively (regime denoted with red line).”  (Line 122-125)

Comment #2

SEM images of studied biochars (Figure 2) look confusing to me. It seems the right-column images are zoomed-in images. It is highly suggested to re-construct the images and use same scale bars. Also it should be cautious of the description in the main text (line 168-171). As to me, I couldn't tell it is porous from the images.

Response #2

Thanks to the reviewer for the question.

SEM images were updated and adjusted to uniform scale bars. We also omitted some images remaining only those ones which clearly demonstrate a complex porous structure.  

Comment #3

The authors should be cautious of the discussion and conclusion (line 361-362). Although the experimental data shows more significant difference on the C/N content, it may not be the direct reason. It could be a comprehensive results from many physical properties.  The micropore size match to the Li+, the Li+ conductivity, etc. may matter.

Response #3

Thanks to the reviewer for the question.

Yes, we agree with your comment. We revised the statement in conclusion: “Among the biochars, those derived from aquatic plants showed the highest capacity, likely due to their composition containing a higher proportion of carbon and nitrogen, as well as optimal particle size and porous structure.” (Line 364-366)